# Influence of Season and Food Type on Bacterial and Entero-Toxigenic Prevalence of *Staphylococcus aureus*

**DOI:** 10.3390/toxins14100671

**Published:** 2022-09-27

**Authors:** Daniela Manila Bianchi, Cristiana Maurella, Christian Lenzi, Massimo Fornasiero, Antonio Barbaro, Lucia Decastelli

**Affiliations:** 1Istituto Zooprofilattico Sperimentale del Piemonte Liguria e Valle d’Aosta, 10154 Turin, Italy; 2National Reference Laboratory for Coaugulase-Positive Staphylococci including *S. aureus*, Istituto Zooprofilattico Sperimentale del Piemonte Liguria e Valle d’Aosta, 10154 Turin, Italy; 3Department of Mathematics, Physics and Natural Sciences, University of Eastern Piedmont Amedeo Avogadro, 13100 Vercelli, Italy

**Keywords:** coagulase-positive staphylococci, staphylococcal enterotoxins, food safety, *S. aureus*

## Abstract

*Staphylococcus (S.) aureus* is a coagulase-positive pathogen of interest for human health and food safety in particular. It can survive in a wide environmental temperature range (7–48 °C, optimum 37 °C). Its enterotoxins are thermostable, which increases the risk of potential contamination in a variety of food products. Here we investigated the influence of seasonality and food type on bacterial count and presence of *S. aureus* enterotoxins. To do this, we analyzed 3604 food samples collected over a 5-year period (2016–2020). Ordinal logistic regression showed an influence of both seasonality and food type on the bacterial count. Regarding bacterial counts, winter was found to be the season with the highest risk, while with regards to enterotoxin production, the highest risk was found in autumn, specifically in October. The risk of contamination with *S. aureus* was greatest for dairy products. Our findings may inform food epidemiologists about foodborne illness prevention and risk to human health.

## 1. Introduction

*Staphylococcus aureus* is the most common species of coagulase-positive staphylococci (CPS). This opportunistic pathogen is widely distributed in nature and has a major impact on human health, making it one of the world’s most common pathogens [1]. Long known as a commensal microorganism residing on the skin and in mucosa, it has recently been recognized as a facultative intracellular pathogen. As such, it plays a critical role in recurrent infection and chronic illness [2].

The pathogenicity of *S. aureus* is attributable to its ability to secrete more than 20 types of toxins [3,4] and invasive enzymes, leading to moderate or potentially fatal illness [5]. The risk of infection is relatively high in humans: an estimated 50–60% are intermittently or permanently colonized with *S. aureus* [6]. While sensitive, rapid, and specific testing can aid in diagnosis and treatment [7], many strains have developed antibiotic resistance. For example, methicillin-resistant strains of *S. aureus* (MRSA) are common in clinical settings, where antibiotic-resistant forms continue to pose a serious problem for hospital-, livestock-, and community-associated infection. The World Health Organization (WHO) has classified them as high-priority health issues [8].

Foodborne disease (FBD) of infectious or toxic nature results from the consumption of contaminated food or water. Many different pathogens are implicated in foodborne infection, one of which is *S. aureus* [9]. *S. aureus* is a common cause of FBD and staphylococcal food poisoning (SFP). SFP occurs after the accidental ingestion of superantigens (e.g., *S. aureus* enterotoxins, SEs) pre-formed in food by *S. aureus* enterotoxigenic strains. Symptoms have a rapid onset (2–8 h): nausea, abdominal cramping, violent vomiting, and diarrhea [10,11] typically resolve within 24 to 48 h after onset.

Staphylococcal contamination of food products occurs through respiratory secretions or via manual contact by individuals carrying the pathogen in the nose or on the hands. Dust, air, and food contact surfaces can also be vehicles of contamination of food items [12]. Due to its biological characteristics, *S. aureus* contamination results mainly from incorrect handling of processed or cooked foods or from storage conditions that allow staphylococcal growth and enterotoxin production. Various food types are subject to *S. aureus* contamination, with products of animal origin frequently involved in cases of SFP. The potential presence of staphylococcal toxins has been reported in meat, ham, eggs, cheese, milk, and dairy items, as well as salads, bakery products, and sandwiches [13,14,15]. In 2009 the European Food Safety Authority (EFSA) stated that the two main vehicles in documented food poisoning outbreaks caused by *S. aureus* toxins were cheese and buffet/mixed meals [16].

Food microbiological safety in Europe is regulated by Regulation (EC) 2073/2005 [17] based on microbiological criteria for foodstuffs. The Regulation sets two types of microbiological criteria with which food businesses must comply, as listed in Annex I: food safety criteria and process hygiene criteria. If test results for either type of criteria are unsatisfactory, food businesses must take specific action in compliance with the Regulation.

The aims of this study were: to determine seasonal changes in CPS concentration and the occurrence of *S. aureus* enterotoxins (SEs) in food product samples from supermarkets, factories, and producers in Italy. The samples were collected and analyzed over a five-year period (2016–2020). The study was designed to evaluate whether the presence of enterotoxins in food products is influenced by food type or seasonality.

## 2. Results

### 2.1. Food Sample Data Set

In our study, we considered a total of 3604 food samples analysed over a 5-year period (2016–2020). Dairy products and ready-to-eat foods made up the bulk of the 3604 food samples delivered to our laboratory for analysis (Table 1). 

### 2.2. Enumeration of Coagulase-Positive Staphylococci

Based on our categorization, most of the samples with a score > 0 (*n* = 3255, 90%) had a score of 1 (10–100 CFU/g) (Table 2). By analyzing the effect of the season of the year, we found out that winter was the season at higher risk for detecting the presence of *S. aureus*, while spring is the season having a sort of protective effect against the presence of a high number of bacteria (Table 2, Table 3 and Table 4; Figure 1 and Figure 2). With regard to the food category, as expected, dairy products showed the highest risk to be contaminated compared to the ready-to-eat category (Table 5 and Table 6).

The winter season was associated with a higher risk than summer; the risk level for autumn and spring was the same as for summer (Table 3).

Figure 1 shows the probability of scores stratified by season. Winter was more likely to have a score of 3 or 4.

Logistic regression with month as the independent variable showed a lower risk for April compared to January (Table 4 and Figure 2).

A higher risk of contamination was associated with dairy products, whereas a lower risk was associated with bakery products (Table 5).

### 2.3. Prevalence of Staphylococcal Enterotoxins

A total of 52/826 samples tested positive for enterotoxins (Table 6). 

Meat, seafood, and bivalve shellfish were excluded from further analysis because of the paucity of samples and the absence of positive results. A higher risk was associated with autumn compared to winter (Table 7). The presence of toxins was highest for October and for dairy products (Table 8 and Table 9, respectively, Figure 3).

## 3. Discussion

Coagulase-positive staphylococci (CPS), and *S. aureus,* in particular, are pathogenic microorganisms of great interest to the food industry. *S. aureus* food poisoning can occur when a minimum of 18 µg SE is present in 100 g of ingested food [1]. The presence of heat-stable enterotoxins in food products can have serious consequences for public health. In *S. aureus* ecology, both seasonality and food type can influence bacterial count and the presence of enterotoxins [18,19,20].

Among the 52 food samples positive for the presence of staphylococcal enterotoxins, 44 were dairy products: 22 were declared to be made of raw milk and no declaration was available for the other 22 samples. Raw milk cheese can be generally considered riskier than pasteurized products because of the potential presence of a higher concentration of bacterial population carrying toxins genes and with an active metabolism to synthesize and secrete toxic proteins. This kind of food should be generally consumed very carefully by young or elderly people.

We found considerable differences in bacterial count and enterotoxin prevalence. Dairy products were most often contaminated with *S. aureus* or staphylococcal enterotoxins: 88.57% of the samples (1550/1750) had a bacterial count score of ≥1 (>10 CFU/g to ≤100 CFU/g). Dairy products, milk, and related foods are at high risk of bacterial contamination, particularly for CPS [21,22].

Analysis of bacterial count of *S. aureus* showed seasonal differences. Compared to the other months, April was most likely to have a score of 0. Although there was no obvious match between the other seasons, the risk of contamination was greater during winter than in summer. Our findings are shared by those of Sentitula and colleagues [23], who based their hypothesis on PCR findings that showed that pathogens (i.e., staphylococci) remain dormant in the udder and then proliferate under suitable climatic conditions. Unlike other studies [18,24] that analyzed the seasonality of *S. aureus* contamination in foods, our dataset was composed of a variety of food types of different origins. CPS thrives at an optimum of 37 °C (range, 7–48 °C); however, the scientific evidence comes largely from dairy products and products of animal origin. The diversity of the food matrices we examined may explain why the incidence of contamination was higher for the colder (winter) compared to the warmer season (spring or summer).

In our dataset, the percentage of samples that tested positive for toxins compared to the total was 6.71% (52 vs. 774), and is consistent with similar studies [25]. We found a marked difference between seasons for the presence of toxins, with autumn being the season most at risk of contamination. A plausible explanation is that there is a greater release of toxins by *S. aureus* in autumn after the warmth of summer, ensuring a higher incidence of toxins in food. A previous study [26] on UHT milk documented a higher production of staphylococcal enterotoxins when the milk was exposed to temperatures between 37 and 42 °C or to fluctuating temperatures.

The European Food Safety approach to protect consumers’ health and guarantee a high safety level in foods, define actions to be undertaken when hygiene or safety criteria are not satisfied. In the frame of Regulation (EC) 2073/2005 [17] coagulase-positive staphylococci hygiene criteria vary from 10 to 10^5^ ufc/g or ufc/mL according to the different food matrices; staphylococcal enterotoxins presence can be investigated on the choice of a competent authority in high-risk foodstuff or must be investigated if CPS concentration is higher than 10^5^ ufc/g or ufc/mL in foods. If any of the hygiene criteria results are uncompliant, the competent authority provides additional investigation on hygiene and good manufacturing practices, own-check analysis results, and cleaning and disinfection protocols. Furthermore, if safety criteria are not satisfied the activation of withdrawal or recall of food products ensures that unsafe products are no longer available on the market.

## 4. Conclusions

Our findings are confirmed by previous studies. They show that seasonality and food type can influence the bacterial count and the presence of *S. aureus* enterotoxins. The data may provide useful information for the food production sector and for public health prevention initiatives.

## 5. Materials and Methods

### 5.1. Food Samples

Overall, 3604 food samples were grouped by food category (Table 1). The survey included samples collected according to the official national plan for food safety by Public Health Services between January 2016 and December 2020 at different points in the food chain, from primary production to retail or catering, in northern Italy. Analysis was performed according to five sampling plans: (1) official food control activities (national plans); (2) criteria verification in mass catering; (3) consultancy activities; (4) official control in the food milk sector, and (5) monitoring and audits by control agencies.

### 5.2. Enumeration of CPS

A total of 3604 analyses for CPS enumeration were performed according to ISO 6888-2:1999 using rabbit plasma fibrinogen agar medium. Test portion, initial suspension, and dilution were performed under aseptic conditions according to ISO 6887-1. Prepared Petri dishes were incubated at 37 °C for 18 h to 24 h. The staphylococci formed black, grey, or small white colonies surrounded by a halo of precipitation, indicating coagulase activity. At the beginning of incubation, proteus colonies may appear like CPS colonies. After incubation for 24 h or 48 h, they appear to spread and are more or less brownish in color, which distinguishes them from staphylococci. Results are expressed as N of CPS/milliliter or per gram of product as weighted.

Some tests were performed using TEMPO^®^: this automated system combines a card with an adapted medium to ensure rapid enumeration of quality indicators. The TEMPO^®^ STA test consists of a vial of culture medium and a card, which are specific to the test. The culture medium is inoculated with the sample to be tested and homogeneously transferred by the TEMPO^®^ Filler into the card containing 48 wells of three different volumes. The TEMPO^®^ Filler seals the card to prevent contamination during subsequent handling. The culture medium contains a fluorescent pH indicator which, when pH is neutral, emits a signal detected by the TEMPO^®^ Reader. During incubation, the CPS present in the card assimilates the nutrients in the culture medium, thus lowering the pH and the fluorescent signal extinguishes. Depending on the number and size of positive wells, the TEMPO^®^ system deduces the number of CPS present in the original sample calculated by the Most Probable Number (MPN) method. Card reading, interpretation, and reporting are managed by the TEMPO^®^ system after incubation for 24 h at 35 °C. The TEMPO^®^ STA method enumerates CPS. Both methods are fully validated and accredited according to ISO 17025: 2017 [27] at the Food Safety Laboratory, Istituto Zooprofilattico Sperimentale del Piemonte Liguria e Valle d’Aosta.

### 5.3. Qualitative Detection of Staphylococcal Enterotoxins

In total, 826 analyses were performed for the qualitative detection of staphylococcal enterotoxins. The protocol proceeds by the following steps according to ISO 19020:2017: extraction, the concentration of the extract, and recovery of the concentrated extract [28]. For the detection of staphylococcal enterotoxins (SEA to SEE) commercially available immunoenzyme detection kits were used: VIDAS® STAPH ENTEROTOXIN II (SET2) (BioMérieux, France) and RIDASCREEN® SET Total (R-Biopharm, Germany). The tests were performed according to the manufacturer’s instructions. All diagnostic protocols are fully validated and accredited at the Food Safety Laboratory, Istituto Zooprofilattico Sperimentale del Piemonte Liguria e Valle d’Aosta.

### 5.4. Statistical Analysis

Data in the Laboratory Information Management System (LIMS) of our Institute and covering a five-year period (2016–2020) were extracted and analyzed. Only data for *S. aureus* count were considered. The bacterial count was scored as follows:

Score 0 CPS < 10 CFU/g or mL

Score 1 >10 and ≤100 CFU/g or mL

Score 2 >100 and ≤1000 CFU/g or mL

Score 3 >1000 and ≤10,000 CFU/g or mL

Score 4 >10,000 and ≤100,000 CFU/g or mL

Score 5 >1,000,000 CFU/g or Ml

The food matrixes were grouped into 26 categories; after data cleaning (all categories with less than 15 records were excluded from analysis) 11 categories remained for analysis (Table 1). The variable season was created by grouping the months January to March (winter), April to June (spring), July to September (summer), and October to December (autumn).

## Figures and Tables

**Figure 1 toxins-14-00671-f001:**
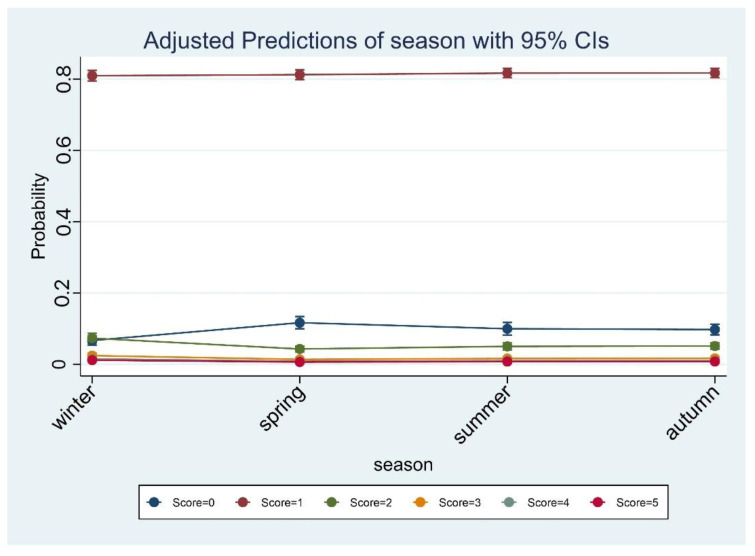
Predicted score probability by season: the probability to detect a very low bacterial count is higher in spring compared to other seasons; the higher scores are almost equally disturbed along the seasons.

**Figure 2 toxins-14-00671-f002:**
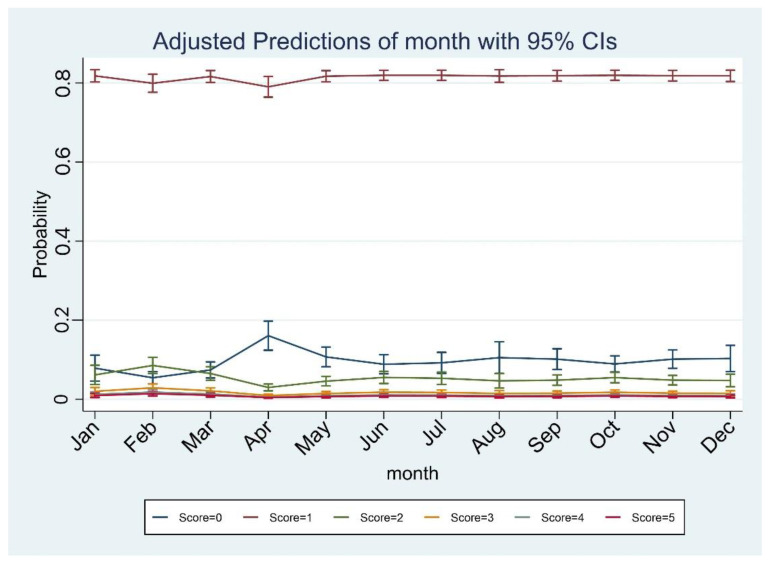
Predicted score probability by month: April is the month at lower risk to report high bacterial count, whereas the higher scores are almost equally distributed throughout the year.

**Figure 3 toxins-14-00671-f003:**
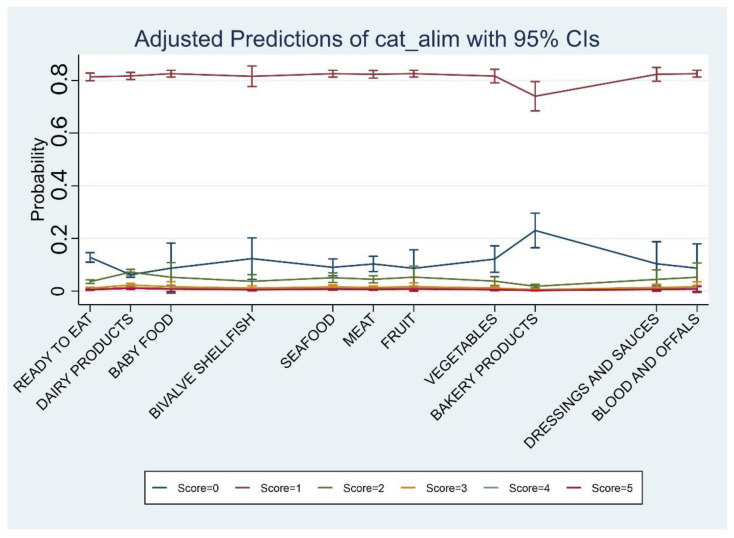
Predicted score probability by food category: bakery products have the highest probability to have a bacterial count equal to zero, whereas dairy products have the highest probability to report a bacterial count score of 4.

**Table 1 toxins-14-00671-t001:** Food matrix categories.

Food Matrix Category	No. of Food Samples
Dairy products	1750
Ready-to-eat	1021
Meat	269
Seafood	168
Bakery products	143
Vegetables	111
Bivalve shellfish	45
Fruit	32
Dressing and sauce	30
Offal	18
Baby food	17
Total	3604

**Table 2 toxins-14-00671-t002:** Food category and score based on the concentration of CPS.

Food Category	Score ^1^						
	0	1	2	3	4	5	Total
Ready-to-eat	81	934	4	1	1	0	1021
Dairy products	200	1254	173	59	35	29	1750
Baby food	0	17	0	0	0	0	17
Bivalve shellfish	3	42	0	0	0	0	45
Seafood	4	161	3	0	0	0	168
Meat	19	239	10	0	1	0	269
Fruit	1	30	0	1	0	0	32
Vegetables	8	102	1	0	0	0	111
Bakery products	32	108	1	1	1	0	143
Dressing and sauce	1	29	0	0	0	0	30
Offal	0	18	0	0	0	0	18
Total	349	2934	192	62	38	29	3604

^1^ Bacterial count of CPS was scored as follows: <10 CFU/g (score 0); >10 and ≤100 CFU/g (score 1); >100 and ≤1000 CFU/g (score 2); >1000 and ≤10,000 CFU/g (score 3); >10,000 and ≤100,000 CFU/g (score 4); >100,000 CFU/g (score 5).

**Table 3 toxins-14-00671-t003:** Ordered logistic regression. Season was the independent variable. Summer was the baseline season.

Season	Odds Ratio	SE	z	P > z	[95% CI]	
Winter	1.550617	0.2050891	3.32	0.001	1.196526	2.009495
Spring	0.838304	0.1035448	−1.43	0.153	0.6580572	1.067922
Autumn	1.024533	0.1255292	0.2	0.843	0.8058123	1.302621

**Table 4 toxins-14-00671-t004:** Ordered logistic regression. Month was the independent variable and January was the baseline season.

Month	Odds Ratio	SE.	z	P > z	[95% CI]	
Feb.	1.488279	0.396517	1.49	0.136	0.8828801	2.508806
Mar.	1.067676	0.2869237	0.24	0.807	0.6305081	1.807959
Apr.	0.4455917	0.1188568	−3.03	0.002	0.2641728	0.751599
May	0.7120983	0.1858644	−1.3	0.193	0.4269414	1.187713
Jun.	0.8798756	0.238019	−0.47	0.636	0.5177963	1.495146
Jul.	0.8420793	0.2323898	−0.62	0.533	0.4902818	1.446306
Aug.	0.726113	0.2267379	−1.02	0.305	0.3937366	1.339068
Sep.	0.7568601	0.2026194	−1.04	0.298	0.4478577	1.279061
Oct.	0.871531	0.2245791	−0.53	0.594	0.5259473	1.444187
Nov.	0.7568948	0.1963986	−1.07	0.283	0.4551623	1.258649
Dec.	0.7431403	0.2150982	−1.03	0.305	0.4214003	1.310529

**Table 5 toxins-14-00671-t005:** Ordered logistic regression. The food category was the independent variable. Ready-to-eat food was the baseline.

Food Category	Odds Ratio	SE	z	P > z	[95% CI]	
Dairy products	2.211108	0.2306706	7.61	0	1.802228	2.712754
Baby food	1.538176	0.9432477	0.7	0.483	0.4624111	5.116629
Bivalve shellfish	1.042982	0.395357	0.11	0.912	0.4961547	2.192486
Seafood	1.479835	0.3115035	1.86	0.063	0.9795724	2.23558
Meat	1.277431	0.2229661	1.4	0.161	0.9073322	1.798491
Fruit	1.549222	0.7103848	0.95	0.34	0.6306684	3.805626
Vegetables	1.060112	0.2648661	0.23	0.815	0.649653	1.729905
Bakery products	0.4904488	0.1005903	−3.47	0.001	0.3281055	0.733118
Dressing and sauce	1.267418	0.5898284	0.51	0.611	0.5090837	3.15537
Blood and offal	1.538176	0.9171231	0.72	0.47	0.478063	4.949109

**Table 6 toxins-14-00671-t006:** Food categories.

Food Category	Negative	Positive	Total
Ready-to-eat	161	5	166
Dairy products	520	44	564
Bivalve shellfish	13	0	13
Seafood	20	0	20
Meat	27	0	27
Vegetables	14	2	16
Bakery products	19	1	20
Total	774	52	826

**Table 7 toxins-14-00671-t007:** Logistic regression: the highest risk was noted for autumn.

Season	Odds Ratio	SE	z	P > z	[95% CI]	
Spring	1.3832	0.6716738	0.67	0.504	0.5340068	3.582807
Summer	2.506667	1.192371	1.93	0.053	0.9867269	6.367899
Autumn	2.513723	1.140172	2.03	0.042	1.033309	6.115114

**Table 8 toxins-14-00671-t008:** Logistic regression: the highest risk was noted for October.

Month	Odds Ratio	SE	z	P > z	[95% CI]	
Feb.	0.7424242	0.6950569	−0.32	0.75	0.1185108	4.650999
Mar.	0.3438597	0.3196726	−1.15	0.251	0.0555972	2.126715
Apr.	1.384181	1.045667	0.43	0.667	0.3148912	6.084503
May	0.4375	0.3649923	−0.99	0.322	0.0852801	2.244442
Jun.	1.053763	0.8296475	0.07	0.947	0.2252041	4.930715
Jul.	2.227273	1.641449	1.09	0.277	0.5253603	9.442556
Aug.	0.6405229	0.5985889	−0.48	0.634	0.1025793	3.999534
Sep.	1.781818	1.307653	0.79	0.431	0.4228416	7.508428
Oct.	3.951613	2.610869	2.08	0.038	1.082374	14.42685
Nov.	0.3888889	0.3618227	−1.02	0.31	0.0627866	2.408709
Dec.	0.5730994	0.5349463	−0.6	0.551	0.0919805	3.570789

**Table 9 toxins-14-00671-t009:** Logistic regression: dairy products were associated with the highest risk.

Food Category	Odds Ratio	SE	z	P > z	[95% CI]	
Dairy products	2.724615	1.309125	2.09	0.037	1.062477	6.987004
Vegetables	4.6	4.056459	1.73	0.084	0.8168367	25.90481
Bakery products	1.694737	1.901464	0.47	0.638	0.1879619	15.2804

## Data Availability

The data presented in this study are available on request from the corresponding author.

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
