# Peer review of "Influence of Season and Food Type on Bacterial and Entero-Toxigenic Prevalence of Staphylococcus aureus"

_toxins, 2022, doi:10.3390/toxins14100671_

Round 1
Reviewer 1 Report
The authors examined the effects of seasonality and type of food on
the number of bacteria and the presence of S. aureus enterotoxins over the period of 5 years (2016-2020). Charts and tables are made correctly and legibly.
Food safety research is very important and should be published in order to make producers and consumers aware of the scale of the problem of food contamination. It should be noted, however, that these studies show large gaps in the safety system of food production, especially dairy products.
The authors write about control from primary production to hadel and catering. I wish I could have found more detailed information on dairy products: was it raw milk? or maybe pasteurized dairy products? After all, it is well known that the cause of most inflammation of the udder in cows is S. aureus, which is then in the milk. After all, at the production stage, it is mandatory to control the total number of microbes and the number of somatic cells, which are indicators of mastitis in cows. Exceeding these rates eliminates raw milk from sale, and cows should be treated with antibiotics. In my opinion, the detected contamination of dairy products indicates that raw milk and cow health controls are not very effective. Such high contamination of dairy products requires frequent interventions and controls during the production cycle. On the other hand, the authors do not write what products were included in the group "dairy products"? I also do not know if this group included sheep and goat milk products?
Certainly, the authors should supplement the applications with information on the actions that must be taken to protect the consumer from consuming toxic products. Readers should also be informed about the consequences that threaten the producers of toxic food. After all, dairy products are consumed mainly by children and the elderly - this is an additional argument for introducing more intensive production controls.
Author Response
Answer in file atteched. thank you very much.

Author Response
Answers in file. thank you very much.

Reviewer 3 Report
Dear all,
I have thoroughly read the article entitled Influence of season and food type on bacterial and enterotoxigenic prevalence of Staphylococcus aureus .
The manuscript sounds good scientifically and written in a professional way.
However, the introduction doesnot support the study and is poor with information. It has to be developed and supported with more information. The same with discussion; very little similar studies are illustrated. The authors should includemore studies supporting their study, wether the results are similar or different. One last major comment; please try to introduce the results in a more clear way to be easier understandable.
Other minor comments are included in the pdf file in a tracking mode.
I hereby accept to publish the current Manuscript after modifications according to the previously listed comments.
Best regards

Author Response
Answers in the pdf file . thank you very much.

Round 2
Reviewer 2 Report
As I mentioned in the first round, the manuscript deals with a topic of current interest and after reading the corrections and data added to the article, as well as the authors' response, I can better understand the main line of the work and I understand that it is publishable, in my opinion, in this journal.